# The Seroprevalence and Hidden Burden of Chikungunya Endemicity and Malaria Mono- and Coinfection in Nigeria

**DOI:** 10.3390/ijerph19158896

**Published:** 2022-07-22

**Authors:** Peter Asaga Mac, Philomena E. Airiohuodion, Andrew B. Yako, James K. Makpo, Axel Kroeger

**Affiliations:** 1Institute of Virology, Universitatsklinikum Freiburg, Hermann Herder Strabe, 11, 79104 Freiburg, Germany; 2Special Programme for Research & Training in Tropical Diseases (TDR), World Health Organization, Avenue Appia 20, 1211 Geneva 27, Switzerland; ehidocng@gmail.com or; 3Department of Zoology, Nasarawa State University, Keffi 911019, Nigeria; 66yakoa@gmail.com (A.B.Y.); jamesmakpo4truth@gmail.com (J.K.M.); 4WHO/Uniklinikum Freiburg, Institute of Molecular Diagnostics and Infectious Disease Control, Keffi 961101, Nigeria; 5Centre for Medicine and Society, University of Freiburg, Bismarckallee 22, 79085 Freiburg, Germany; axel.kroeger@zmg.uni-freiburg.de

**Keywords:** chikungunya, coinfection, seroprevalence, Nigeria, endemicity, malaria

## Abstract

Background: Mosquito-borne infections are of global health concern because of their rapid spread and upsurge, which creates a risk for coinfections. Chikungunya, an arbovirus disease transmitted by *Aedes aegypti* or *A.* *albopictus*, and malaria, a parasitic disease transmitted by *Anopheles gambiae*, are prevalent in Nigeria and neighbouring countries, but their burden and possible coinfections are poorly understood. In this study, we investigated the seroprevalence, hidden burden and endemicity of chikungunya and malaria in three regions in Nigeria. Methods: A cross-sectional sero-survey was conducted on 871 participants in three regions of Nigeria. The samples were collected from outpatients employing simple random sampling. All serum sample analyses were performed using CHIKV virus-like particle recomLine Tropical Fever for the presence of arboviral antibody serological marker IgG immunoblot for chikungunya and malaria RDT (Rapid Diagnostic Test) for malaria parasites. Results: The seroprevalences of chikungunya and malaria mono-infection were 64.9% and 27.7%, respectively, while the coinfection seroprevalence was 71.9%. The central (69.5%) and northern (67.0%) regions showed more significant seroprevalences than the southern region (48.0%). The seroprevalence and the hidden burden of chikungunya and malaria infections varied across the three geographical regions. Conclusions: This study highlighted an unexpectedly high seroprevalence and hidden endemicity of chikungunya and a less surprising high malaria endemicity in three regions of Nigeria.

## 1. Background

There are over 700,000 deaths annually due to vector-borne diseases, accounting for more than 17% of all infectious diseases [1,2,3,4]. Mosquito-borne infections are currently one of the leading global health concerns due to their continued spread and upsurge, thus posing a threat to new regions with new pathogens [3,4,5]. As a result, endemic pathogens can be transmitted by imported vectors, or newly introduced pathogens can be transmitted by local vector populations. The chikungunya virus is transmitted from mosquitoes to humans through the bite of infected mosquitoes of the genus *Aedes*. *Aedes aegypti* and *Aedes albopictus* are the primary vectors of the chikungunya virus (CHIKV), but it is also maintained in sylvatic cycles involving primates and forest dwelling *Aedes* species. Chikungunya is one of the most common vector-borne diseases [5,6,7,8], and its related symptoms (fever, headaches, rash, muscular pains, joint pains) are similar to those of malaria and other febrile illnesses. The possibility of misdiagnosis in the case of these AFIs is therefore high, particularly in sub-Saharan Africa [4,5,6,9].

On the other hand, malaria is a major cause of fever in all age groups, contributing to significant infant mortality in Nigeria [7,8,10]. *Anopheles gambiae* is the principal vector of malaria in Nigeria and *An. arabiensis* and *An*. *funestus* are secondary [10]. Secondary vectors often have relatively little contact with humans and may be less susceptible to residual insecticide spraying or insecticide-impregnated bednets than primary vectors [7]. There is an increased risk of chikungunya and malaria occurring concurrently and causing malaria and nonmalaria febrile illnesses [11,12,13]. Chikungunya and malaria can also coexist in areas where the two vectors are present [4,8,14,15,16,17,18,19,20]; therefore, super- or coinfection is a possibility.

Malaria and chikungunya coinfections are among the most common mosquito-borne infections in Nigeria [18,19,20], in addition to dengue/chikungunya and malaria/dengue/chikungunya [1,4,5,6,7,8,16,17,18]. Studies have shown that the prevalence of coinfections of AFIs, such as malaria and chikungunya, can lead to severe disease and fatal outcomes [4,5,6,7,8,16,18,21,22,23,24,25,26,27,28] with serious public health implications [7,10,14,15,29]. To our knowledge, there is insufficient evidence on the seroprevalence, hidden burden (morbidity associated with misdiagnosis) and endemicity of chikungunya and malaria coinfections in Nigeria and many other parts of western Africa. The aim of this study was to investigate the seroprevalence and hidden burden of chikungunya endemicity and to determine its coinfection (serologically positive cases of chikungunya and malaria at the time of sampling or study) with malaria.

## 2. Methods

### 2.1. Study Design and Site

A cross-sectional study was conducted in three university teaching hospitals in Nigeria, namely, the Federal Medical Centre, Keffi located in Nasarawa State, Central Nigeria Abia State University Teaching Hospital, Aba located in Abia State, Southern Nigeria and Baru-Diko Teaching Hospital, Kaduna in Kaduna State, located in Northern Nigeria (Figure 1). These facilities are among Nigeria’s largest outpatient clinics and maternity hospitals, serving approximately 10–26 million people each year. The three states or regions have a combined population of over 30 million inhabitants. The average annual temperatures range from 21 °C to 27 °C, while in the interior lowlands, temperatures are generally above 27 °C. The mean annual precipitation is 1165.0 mm [10]. It rains throughout the year in most parts of southern and central Nigeria, with most rainfall occurring between April and October and minimal rainfall occurring between November and March in the north. The main occupation of the inhabitants of the three states or regions is farming at both the commercial and subsistence levels.

### 2.2. Study Population

The study population were all outpatients, among whom were pregnant women enrolled for antenatal care and patients presenting with illness at the rapid-access healthcare units of the tertiary hospitals between January 2021 and November 2021. These hospitals were purposefully selected to reflect the diversity (in terms of different cultures, religions, ethnicities, topographical and vegetation features, and different human activities) of the three geographical regions. Inclusion criteria were all outpatients within an age range of 0 month–80 years who agreed to participate in the study and signed the consent form, including children whose parents or guardians gave consent, while exclusion criteria were participants who were already undergoing treatment for malaria, those who refused to sign the consent form and seriously ill patients who were hospitalized.

### 2.3. Screening of Study Participants

A structured questionnaire was used to obtain information that included questions on demographics, medical history, vital signs and symptoms, clinical evaluation, data on hospitalisation, and a summary form. All study subjects were screened for malaria and chikungunya-related symptoms (fever, headaches, rashes, joint pain, conjunctivitis, and muscular pain). Detailed protocol information was made available and fully explained to the participants in English and their respective local languages before enrolment. The study participants signed an informed consent form after enrollment. Participants who could not read and write were asked to verbally consent and then to thumb print indicating that they were willing to participate in the study.

### 2.4. Total Number of Samples Collected

A total of 871 samples were collected from the three sites employing a simple random sampling method. Of these, 761 samples were collected from outpatients, and 110 samples were taken from blood banks. The sample size calculation (based on a 40% expected proportion of chikungunya and malaria infections in a total population of 500 thousand patients with a confidence interval of 95% and a *p* value of 0.05) showed a minimum sample size of 384 serum samples, which we increased to 871 samples to be able to analyse subgroups according to regions.

Venous blood was collected (5 mL) from all participants by the principal investigator and his assistants. This was performed throughout the 12-h shift. Additionally, a small number of 110 blood samples together with the clinical history were collected from the blood banks in the 3 hospitals by a local clinical diagnostic laboratory technician (located in the hospital who collected patient blood samples daily). The serum was extracted and screened at the study site in Nigeria for malaria by employing a Malaria Rapid Antigen Detection Test (RDT kit) (SD BIOLINE Malaria Differential P.f/Pan Ag RDT (HRP II + pLDH, Abbott, Chicago, IL, USA) and thereafter shipped on dry ice to the Institute of Virology, Freiburg, Germany. The serum samples were stored at −20 °C in preparation for laboratory analysis for chikungunya infection.

### 2.5. Laboratory Analysis

All serum samples were screened for malaria parasites in the field, employing a Malaria Rapid Antigen test (SD BIOLINE Malaria Differential P.f/Pan Ag RDT (HRP II + pLDH, Abbott) according to the manufacturer’s instructions. In summary, 5 µL of blood sample was transferred into the sample well using the appropriate device included in the kit, and five drops of lysis buffer were added to the buffer well. The results were read visually after 15–20 min. For CHIKV, analyses were performed using the recomLine Tropical Fever for the presence of arboviral antibody serological marker IgG immunoblot (Mikrogen Diagnostik, Neuried, Germany) CHIKV VLP (virus-like particle) according to the manufacturer’s instructions [30,31]. In summary, the recomLine Tropical Fever chikungunya IgG/IgM virus-like particles (VLPs) is a line immunoassay (in contrast to ELISA test systems) that allows the identification of specific antibodies against a single antigen of chikungunya from other alpha viruses.

### 2.6. Statistical Tests

Statistical analysis was performed using SPSS version 28 (IBM, Armonk, NY, USA). Descriptive statistics were employed for the analysis of results and 95% confidence intervals [CI] to identify the sociodemographic and behavioral characteristics of the study population. The results are presented in tables and figures. A Chi-square test of independence and correlations were performed to examine the relationship between the variables and chikungunya–malaria seroprevalence. The results were deemed statistically significant at a *p* value ≤ 0.05.

### 2.7. Ethics Statement

The study protocol was reviewed and approved by the local ethics committee on human research at the Universitatsklinikum, Freiburg, [No. 21-1233] and the local ethics committee on human research at the Tertiary Hospitals and national ethics committee on human research of Nigeria [No KF/REC/02/21].

## 3. Results

### 3.1. The Demographic Profiles and Seroprevalence of Chikungunya and Malaria

A total of 871 participants [male: 29.0% (252/871); female: 71.0% (619/871)] were investigated for the seroprevalence of chikungunya and malaria parasites in three Nigeria regions. The mean age of the study population was 36.7 years (range 0 month to 80 years). In the three profile regions, the seroprevalence of chikungunya was 64.9% (95% CI (0.63–0.67)) and that of malaria was 35.1% (95% CI (0.32–0.38)). A significant seroprevalence of chikungunya was observed among participants from the central (Nasarawa) region (69.5% (95% CI (0.68–0.70))) and northern (Kaduna) region (67.0% (95% CI (0.65–0.69))), while a slightly lower seroprevalence of 48.0% (95% CI (0.46–0.50)) was revealed among the participants from the southern (Abia) region (Table 1). The difference between these variables was statistically significant (*p* = 0.01). There was a low positive correlation and a statistically significant association between Abia (ABIA), Nasarawa (NAS), and Kaduna (KAD) with respect to chikungunya infection (*p* = 0.03).

The seroprevalence of malaria was slightly higher in the southern region (44.7%, 95% CI (0.33–0.57)) than in the northern (31.5%, 95% CI (0.24–0.39)) and central regions (14.0%, 95% CI (0.4–0.24)). There was no statistically significant difference between malaria infection and the regions (*p* = 0.65). However, there was a negative correlation and a significant association between the regions and malaria infection (*p* = 0.01).

### 3.2. Age-Specific Seroprevalence of Chikungunya and Malaria in the Study Population

Chikungunya infection seroprevalences ranged from 36.7% to 75.0%, and malaria ranged from 10.0% to 65.0% in the study population. It was observed in Abia and Kaduna that the median age for chikungunya-infected persons was almost the same as that of noninfected persons, but it was slightly higher in Nasarawa. The mean age of malaria-infected individuals was slightly lower in Kaduna but almost the same in Abia and Nasarawa (Figure 2).

### 3.3. Sex-Specific Seroprevalence of Chikungunya and Malaria

In the present study, a lower chikungunya seroprevalence of 64.3% (95% CI (0.61–0.67)) was observed among males than among females (65.1% (95% CI (0.62–0.68))). Similarly, malaria seroprevalence was also higher among females (29.9%, 95% CI (0.27–0.33)) than males (22.6%, 95% CI (0.19–0.25)) (Table 1). There was a statistically significant difference between the sexes (males and females) and malaria (*p* = 0.03); however, the difference between the sexes (females and males) and chikungunya was not statistically significant (*p* = 0.82).

### 3.4. Place Specific Seroprevalence of Chikungunya and Malaria

The seroprevalence of chikungunya was highest (70.8% (95% CI (0.69–0.73))) among slum dwellers, compared to 68.8% (95% CI (0.69–0.72)) rural and 61.1% (95% CI (0.59–0.63)) urban dwellers. There was a statistically significant association between place of domicile and chikungunya infection (*p* = 0.01).

A markedly higher seroprevalence of malaria was evident in slums or informal participants (34.9%, 95% CI 0.20–0.50) than in rural (32.2%, 95% CI 0.22–0.42) and urban participants (24.1%, 95% CI 0.12–0.26). The results showed a significant association between place of residence and malaria (*p* = 0.01), with a negative correlation (Table 1).

### 3.5. Pregnant and Nonpregnant Female Seroprevalence of Chikungunya and Malaria in the Study Population

A higher chikungunya seroprevalence rate of 66.3% (95% CI (0.63–0.67)) was obtained in nonpregnant women than in their pregnant counterparts (60.9%, 95% CI (0.60–0.64)). The correlation was negative, and the difference was not statistically significant (*p* = 0.14). In contrast, a significant malaria seroprevalence of 33.0% (95% CI (0.22–0.44)) was recorded among pregnant women compared with nonpregnant women (25.9%; 95% CI (0.19–0.33)). There was a statistically significant association between pregnancy status and malaria (*p* = 0.04), with a negative correlation (Table 1).

### 3.6. Seroprevalence of Chikungunya–Malaria Coinfection in the Study Population

A combined seroprevalence of 71.9% (95% CI (0.70–0.74)) chikungunya–malaria coinfection was observed among the study participants in the three regions. A subgroup analysis of the demographic groupings revealed an overall coinfection seroprevalence of 68.4% among all age groups, which was closely followed by the domicile group (66.9%) and the male and female groups (65.1%). The lowest prevalence was observed among the pregnant and nonpregnant women groups (62.6%). The relationship between these demographic variables and chikungunya–malaria coinfection was statistically significant (Table 2).

## 4. Discussion

Chikungunya and malaria coinfections could result in a significant public health impact and fatal outcomes through aggregates of various factors and inaccurate and misdiagnosis. In the present study, we investigated the seroprevalence and burden of chikungunya hidden endemicity and malaria mono- and coinfection among subpopulation groups in three regions of Nigeria. Our findings showed that the study population has a high seroprevalence and a hidden burden of chikungunya (64.9%) and malaria mono- and coinfection (71.9%) across various sociodemographic groups. The seroprevalence and heterogeneity of infections across the various geographical regions may be explained by differences in vegetation, human population, climate change (increasing the reproductive activities and shortening the extrinsic cycle of chikungunya and malaria in the vector), vector adaptations, variations in temperatures and humidity, changes in habitats and microclimates, and unplanned urbanisation in the various regions that favour transmission dynamics of the mosquito-borne vectors of these diseases [15,16]. These findings are consistent with other seroprevalence studies conducted in other parts of Nigeria, west Africa, and the rest of the world [1,2,3,4,5,6,7,7,8,10,16,18,25]. Misdiagnosis and inadequate or underestimated testing capacities of Nigerian laboratories to diagnose these mosquito-borne coinfections make quantifying the hidden burden and endemicity difficult [1,2,3,4].

Chikungunya infection was more prevalent in older age groups than malaria, which was more common in younger age groups [32]. It is also evident in our findings that the seroprevalence of chikungunya seems to increase with age [32], while malaria prevalence appears to decrease with age. Chikungunya IgG antibodies last for many years following infection with the chikungunya virus (IgG is detectable 7–10 days after CHIKV infection, but remains detectable for several years, providing long-term immunity). In children, malaria and other arboviral infections are more prevalent due to loss or weaning-off of maternal immunity [32].

The high seroprevalences among the population could also be attributed to past infection, reinfection or ongoing transmission or increased vector exposure in relation to socioeconomic activities close to mosquito breeding habitats. Furthermore, older people maintain sedentary lifestyles, and because they sit for long periods in unscreened places, they are increasingly exposed to *Aedes* mosquito bites (day feeding activity of *Aedes aegypti*), while children are more vulnerable to malaria than chikungunya. This age bias has been previously reported [22] and may be attributed to sociocultural habits and behaviours.

The seroprevalences of chikungunya and malaria mono- and coinfections in female participants were higher and more discrete than those in their male counterparts. These differences could possibly be due to the number of females recruited in this study compared to males (more females were recruited). Additionally, females in Nigeria engage in outdoor activities such as trading, fetching water from ponds and streams, and gathering wood for fuel and farming, just as much as their male counterparts, thereby exposing them to the breeding habitats of *Anopheles* and *Aedes* mosquitoes [7,10,14].

There was a higher seroprevalence of chikungunya and malaria in slums than in rural and urban settlements in the present study. This could be explained by the fact that slums are informal settlements located in urban areas or cities across the three profiled regions [1,2,3]. As a result, several factors, including poor housing favouring mosquito breeding habitats, uncontrolled urbanisation, cultural and behavioural practices resulting in poor sewage and drainage infrastructures, and unhygienic conditions of the environment, were compounded by the unwholesome location of waste disposal dumpsites close to various homes in slums and urban areas in the study regions. Rural–urban migration, expansion of agricultural activities to sylvatic areas in rural settings, and political and conflict fatigue may also be the drivers of these high seroprevalences in the three regions [8,13,17].

The prevalence of malaria was higher among pregnant women, whereas chikungunya was more prevalent among nonpregnant women. These results could be explained by the differences in immune-physiological state during pregnancy, thus resulting in reduced or weakened immunological responses to mosquito-borne infections [15,24], malaria is notably more severe during pregnancy. It has been shown that arboviral haemorrhagic complications increase maternal mortality and the risk of C-sections and postpartum haemorrhages [17,18,19]. However, these were not investigated in the present study.

It was evident in the present study that there was a high and widespread seroprevalence of chikungunya–malaria coinfection. These findings could be attributed to the spread of chikungunya in various regions in Nigeria where malaria is endemic and could possibly result in an increase in concurrent infection [4,5,6,7,8,14,15,16,29]. It is unclear whether the presence of malaria parasites reactivates or increases sensitivity to chikungunya or whether chikungunya infection increases or reactivates malaria. The high occurrence of malaria and chikungunya coinfection seroprevalence among the study participants could possibly reveal the extent of undiagnosed, misdiagnosed, hidden burden and prevalence of the two mosquito-borne infections. It could also be the lack and limited or underestimated testing capacities of regional health systems, and epidemiological serosurveillance facilities. The high seropositivity could also be explained by the inability to accurately diagnose malaria and arboviral infections and distinguish them from other febrile illnesses [1,2,6]. A few studies from other parts of Africa [7,18,20,22] reported similar findings.

## 5. Limitations

The study population is hospital-based; therefore, the seroprevalence could not be a true reflection of what is obtainable in the general Nigerian population as such, and it could be underestimated. Although we employed a novel molecular diagnostic test that is highly specific but not a seroneutralisation test, it still remains a challenge to diagnose chikungunya and other alpha viruses using serological tests due to cross-reactivity, which complicates the interpretation of the results. The interpretation of IgG antibodies may also be complicated (false positives or negatives) by previous exposure to mosquito-borne arbovirus vaccinations. Furthermore, the obvious clinical presentations of arboviral infection, apart from being subclinical, can mimic other infections, such as bacterial and fungal infections, which are also endemic in Nigeria. Among the participants in the present study, there were more females than males, which may lead to bias and confounding other variables as well as age.

## 6. Conclusions

We were able to show that there is a high seroprevalence and hidden endemicity and burden of chikungunya and malaria mono- and coinfection in Nigeria. Several intrinsic and extrinsic factors were responsible for these high seroprevalences. It is evident that these two infections could go unnoticed, especially when causing fever, and will end up being treated as other common febrile illnesses or bacterial and fungal infections because of their subclinical presentation. According to our findings, molecular laboratory diagnostics and serosurveillance epidemiological facilities are essential for the accurate and early diagnosis of arboviral mono- and coinfections in Nigeria.

## Figures and Tables

**Figure 1 ijerph-19-08896-f001:**
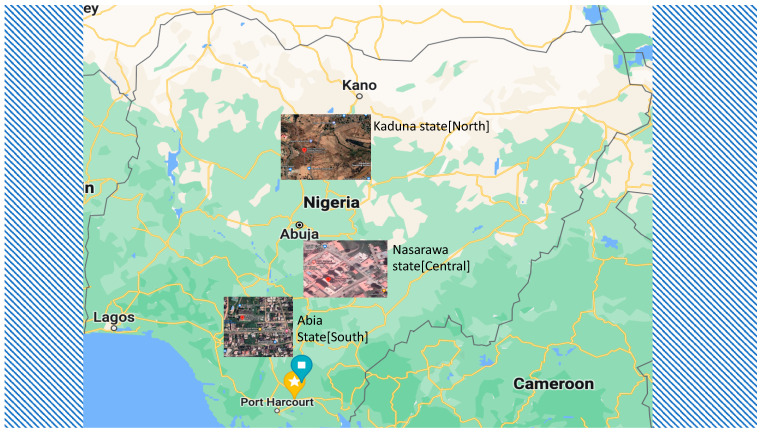
Chikungunya and malaria study sites in Nigeria.

**Figure 2 ijerph-19-08896-f002:**
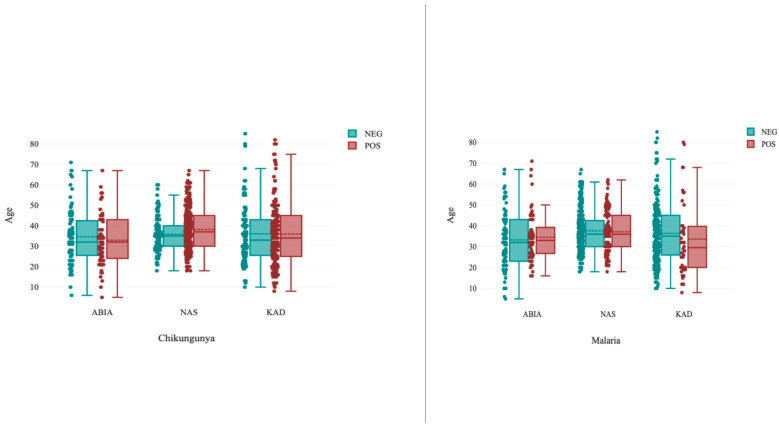
Boxplot of the median age of infected and noninfected persons with chikungunya and malaria in three regions of Nigeria. Age specific seroprevalence of chikungunya and malaria in the three regions. Abia (ABIA; Southern region); Nasarawa (NAS: Central region); Kaduna (KAD; Northen region).

**Table 1 ijerph-19-08896-t001:** Seroprevalence of chikungunya and malaria in the three regions of Nigeria.

	Chikungunya						Malaria					
	Negative	Positive	Total Examined (n)	95% CI			Negative	Positive	Total Examined (n)	95% CI		
Regions												
South (Abia)	79 (52.0%)	73 (48.0%)	152 (100%)	0.46–0.50	*p* = 0.03	R = 0.16	84 (55.3%)	68 (44.7%)	152 (100%)	0.33–0.57	*p* = 0.01	R = 0.25
North (Kaduna)	99 (33%)	201 (67.0%)	300 (100%)	0.65–0.69			287 (68.5%)	132 (31.5%)	419 (100%)	0.24–0.39		
Central (Nasarawa)	128 (30.5%)	291 (69.5%)	419 (100%)	0.68–0.70			258 (86.0%)	42 (14.0%)	300 (100%)	0.4–0.24		
Total (N)	306 (35.1%)	565 (64.9%)	87 (100%)	0.63–0.67			629 (72.3%)	242 (27.7%)	871 (100%)	0.25–0.31		
Sex												
Male	90 (35.7%)	162 (64.3%)	252 (100%)	0.61–0.67	*p* = 0.82	R = 0.01	195 (77.4%)	57 (22.6%)	252 (100%)	0.19–0.25	*p* = 0.03	R = 0.01
Female	216 (34.9%)	403 (65.1%)	619 (100%)	0.62–0.68			434 (70.1%)	185 (29.9%)	619 (100%)	0.27–0.33		
Total (N)	306 (35.1%)	565 (64.9%)	871 (100%)	0.62–0.68			629 (72.3%)	242 (27.7%)	871 (100%)	0.25–0.31		
Domicile												
Urban	197 (38.9%)	310 (61.1%)	507 (100%)	0.59–0.63	*p* = 0.02	R = 0.09	385 (75.9%)	122 (24.1%)	507 (100%)	0.12–0.26	*p* = 0.01	R = 0.1
Rural	78 (30.2%)	180 (69.8%)	258 (100%)	0.68–0.72			175 (67.8%)	83 (32.2%)	258 (100%)	0.22–0.42		
Slum	31 (29.2%)	75 (70.8%)	106 (100%)	0.69–0.73			69 (65.1%)	37 (34.9%)	106 (100%)	0.20–0.50		
Total (N)	306 (35.1%)	565 (64.9%)	871 (100%)	0.63–0.67			629 (72.3%)	242 (27.7%)	871 (100%)	0.25–0.31		
Pregnancy Status												
Pregnant	88 (37.8%)	145 (60.9%)	233 (100%)	0.60–0.64	*p* = 0.32	R = 0.05	154 (67.0%)	76 (33.0%)	230 (100%)	0.22–0.44	*p* = 0.04	R = 0.07
Non-pregnant	218 (34.6%)	420 (66.3%)	638 (100%)	0.63–0.67			475 (74.1.0%)	166 (25.9%)	641 (100%)	0.19–0.33		
Total (N)	306 (35.1%)	565 (64.9%)	871 (100%)	0.63–0.67			629 (49.0%)	242 (27.7%)	871 (100%)	0.25–0.31		

**Table 2 ijerph-19-08896-t002:** Seroprevalence of chikungunya–malaria coinfection in the study population.

Demographic Groups	Chikungunya-Malaria Co-Infection					
	Negative	Positive	Total Examined (n)	95% CI		
Regions (South, North, Central)	20 (9.6%)	189 (90.4%)	209 (100%)	0.88–0.92	*p* = 0.01	R = 0.25
Age	79 (31.6%)	171 (68.4%)	250 (100%)	0.66–0.70		
Sex (Males & Females)	52 (34.9%)	97 (65.1%)	149 (100%)	0.63–0.68		
Place of domicile (Urban, Rural, Slum)	41 (33.1%)	83 (66.9%)	124 (100%)	0.68–0.72		
Pregnancy status (pregnant/nonpregnant)	52 (37.4%)	87 (62.6%)	139 (100%)	0.61–0.65		
Grand Total Examined (N)	244 (28.0%)	627 (71.9%)	871 (100%)	0.70–0.74		

## Data Availability

All data are contained within the manuscript.

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
