# Peer review of "The Seroprevalence and Hidden Burden of Chikungunya Endemicity and Malaria Mono- and Coinfection in Nigeria"

_ijerph, 2022, doi:10.3390/ijerph19158896_

Round 1

Reviewer 1 Report

TITLE

Avoid that the title is gimmicky. I suggest removing the words "high" and "endemicity"

ABSTRACT

I suggest rewriting the methods section so that it is better understood. It should be indicated if the sample was collected from healthy volunteers or if some type of sampling was done.

BACKGROUND

The sentences between lines 43-47 do not add context to the study. I suggest increasing the context regarding the seroprevalence of both diseases in the continent or the country. I suggest considering this reference: Igbasi U, Oyibo W. Seroprevalence of immunoglobulin G and E among out-patients with malaria in Ikorodu Lga, Lagos, Nigeria. Microbes Infect Chemother. 2022; 2: e1376.

METHODS

Support the reason for including such young children in the study (0 months). How frequent is the prevalence in this age? How they have evaluated the bias of decreasing prevalence by having children in the study.

I strongly suggest removing children under 5 years of age from the analysis, due to the high bias they cause, not only in seroprevalence but also in the symptoms and signs evaluated.

Because they have chosen blood bank samples. Prior to the taking, an interrogation is made to define if the donation proceeds. This affects serology results for both diseases and even more so for malaria. How have clinical questions been evaluated in blood bank samples with clinical histories? How much missing data did you have in those 110 samples? How did you choose which samples to obtain from the blood bank? Was it also by simple random sampling?

RESULTS

Avoid repeating the data shown in the tables in the text.

Increase the size of figure 2.

Figure 3 can be described in the text, it is unnecessary.

Figures 4 and 5 can be shown in a table.

DISCUSSION

Consider the discussion in other contexts of the disease. I consider this research relevant in terms of socioeconomic aspects: Olivera MJ, Peña C, Yasnot MF, Padilla J. Socioeconomic determinants for malaria transmission risk in Colombia: An ecological study. Microbes Infect Chemother. 2022; 2: e1339

It is an interesting study, however, the wide age range and the use of blood bank samples lead to serious deficiencies. I suggest removing it from the analysis.

Author Response

   Reviewer 1

Comments and Suggestions for Authors

TITLE

Avoid that the title is gimmicky. I suggest removing the words "high" and "endemicity"

Answer: Sir it is what we saw that made us come up with the title. According to your suggestion, we shall take out the “High”.

New title: The seroprevalence and hidden burden of chikungunya endemicity and malaria mono and coinfection in Nigeria”

ABSTRACT

I suggest rewriting the methods section so that it is better understood. It should be indicated if the sample was collected from healthy volunteers or if some type of sampling was done.

Answer: The abstract method section was rectified regarding the sampling method and from whom the samples were collected.  

BACKGROUND

The sentences between lines 43-47 do not add context to the study. I suggest increasing the context regarding the seroprevalence of both diseases in the continent or the country. I suggest considering this reference: Igbasi U, Oyibo W. Seroprevalence of immunoglobulin G and E among out-patients with malaria in Ikorodu Lga, Lagos, Nigeria. Microbes Infect Chemother. 2022; 2: e1376.

Answer: Rectified with [35]

METHODS

Support the reason for including such young children in the study (0 months). How frequent is the prevalence in this age? How they have evaluated the bias of decreasing prevalence by having children in the study.

Answer: The burden of CHIKV infection is particularly in the very young age groups [arboviral took KIT, 2018]. Most frequently the infection is diagnosed as malaria or other febrile illness.  [WHO arboviral tool KIT, 2018].  

I strongly suggest removing children under 5 years of age from the analysis, due to the high bias they cause, not only in seroprevalence but also in the symptoms and signs evaluated.

Answer: As mentioned above, under-fives are particularly affected by arbovirus diseases and malaria. [WHO, Dengue tool, KIT]. This is why we included them in the current study. And we think we had important findings Therefore we think it is important to include them showing that many infections occur at such a low age.  

Because they have chosen blood bank samples. Prior to the taking, an interrogation is made to define if the donation proceeds.

Answer: Blood donors are nor checked for Antibodies in our target diseases. This was a small additional study that did not change our overall results.

This affects serology results for both diseases and even more so for malaria. How have clinical questions been evaluated in blood bank samples with clinical histories?

Answer: All the blood samples from the blood bank had all the medical data such as date, blood group, genotype, matrix tracking code, donor ID number or product code (Which contain the age and sex, entered in hospital records) expiration information. We had all the information. The samples were selected randomly by the hospital laboratory technician of participating hospitals. 

How much missing data did you have in those 110 samples? How did you choose which samples to obtain from the blood bank? Was it also by simple random sampling?

Answer: All the blood samples from the blood bank had all the medical data such as date, blood group, genotype, matrix tracking code, donor ID number, product code (Which contain the age and sex, entered in hospital records, very confidential) expiration information. We had all the information. It was selected randomly by the hospital laboratory technician located in the hospitals.

RESULTS

Avoid repeating the data shown in the tables in the text.

Answer: In the text we put only the important data without quoting the 95%Cis and p-values. These the reader can find in the text.

Increase the size of figure 2.

Answer: The size has been increased.

Figure 3 can be described in the text, it is unnecessary.

Answer: Taken out

Figures 4 and 5 can be shown in a table.

Answer: Taken out as suggested.

DISCUSSION

Consider the discussion in other contexts of the disease. I consider this research relevant in terms of socioeconomic aspects: Olivera MJ, Peña C, Yasnot MF, Padilla J. Socioeconomic determinants for malaria transmission risk in Colombia: An ecological study. Microbes Infect Chemother. 2022; 2: e1339

Answer: Noted sir and taken into account (see text)

It is an interesting study, however, the wide age range and the use of blood bank samples lead to serious deficiencies. I suggest removing it from the analysis.

Answer: Answer: Various arboviral infections, such as chikungunya, have been reported to be transmitted through blood donation. Blood bank samples are crucial for studies like this. In most blood banks and clinics, arboviruses are rarely screened [Haliya et al [2021]: Seroprevalence of Dengue and Chikungunya antibodies among blood donors in Dar es Salaam and Zanzibar]. Due to inadequate molecular diagnostic tools and maternal immunity, juveniles and neonates bear the hidden burden of misdiagnosis and associated morbidities of arboviruses[WHO, dengue took KIT].  We do not know why it should be ruled out or removed in the current study, sir. For us, it’s the very core part of the substance and content of the study.

Reviewer 2 Report

This nigerian-german cross-sectional study was conducted in a large cohort of outpatients from 3 hospitals in distinct regions of Nigeria. After informed consent, patients’ sera were tested on site for malaria antigen detection (detection of ongoing infection) with a rapid diagnostic test. Sera was screened in Germany for chikungunya virus antibodies with a commercial immunoblot kit (detection of past infection). As reported by the authors, results show endemic-like prevalence of chikungunya virus in Nigeria with significant differences in regions and habitation environments. Authors state that the prevalence of malaria infections is lower in contrast to chikungunya infections with significant differences in habitation environments and pregnancy status.

Due to the rarity of arboviral activity in Africa, seroprevalence or molecular data is always welcome for informed readers. This study is particularly unique for its patient cohort sampled in three sites representing a gradient of climates from arid to tropical. This sample collection could deliver valuable data for epidemiologists, clinicians.

The studies’ objective as stated by the authors was to investigate the seroprevalence of chikungunya and determine coinfection with malaria. Overall, in the current state of the manuscript, I do not agree that presented used methods answer the studies objectives and I do not agree with the current conclusions drawn from presented results.  I also feel there is a misuse of certain terms such as “seroprevalence”, “coinfection” and “burden”. Detailed arguments bellow.

To my understanding, a “coinfection” is a simultaneous infection of two pathogens, malaria and chikungunya virus in this instance. However, the methods used detect ongoing infections of malaria and past infections of chikungunya virus. Subsequent results cannot demonstrate simultaneous infections; thus the use of co-infection with the current results is misguiding to my opinion. The authors should consider screening for CHIKV RNA in the sera to prove co-infection.  

I would also argue that a unique commercial immunoblot result is not sufficient to prove an arboviral infection. In the matter of biological confirmation of arboviral infection, the WHO has established gold standard practices: a positive serological result should be confirmed by a seroneutralisation test (sole confirmatory test available for serological diagnosis) due to cross-reactivity of CHIKV-IgG towards other alphaviruses. Considering those recommendations, the serological evidence present in this study can only indicate a probable CHIKV infection and not a confirmed past CHIKV infection. Limitations concerning cross-reactivity of alphaviruses, prevalence of other alphaviruses in Nigeria, sensitivity and specificity of commercial tests (also, no mention of HRP2-deleted plasmodium in discussion) are all major study limitation which were not discussed. Authors should perform a seroneutralisation assay on a panel of relevant alphaviruses to support their conclusions.

The authors use the term “seroprelavence” for results of chikungunya-IgG detection and malaria antigen detection. However, the term “seroprelavence” is entailed by definition to antibodies detected in a population. Therefore, the direct detection malaria antigen and not the host’s antibodies is “prevalence” rather than “seroprevalence”. The seroprevalence of chikungunya virus and the prevalence of malaria should be treated and compared with more caution. I personally do not believe such data are comparable: on one hand, life long-lasting highly cross-reactive antibodies of an auto-resolutive viral infection are detected and on the other an antigen of chronically infection parasite with complex biological cycle and complex immunological host response.

In addition, the title of the article mentions a “high burden” of chikungunya virus. However, the burden of a disease is defined as the consequences (health, social aspects and cost to society) of a defined disease or a range of harmful diseases with respect to disabilities in a community. The authors do specify that chikungunya and malaria related symptoms were surveyed. Yet, ongoing chikungunya infection were not evaluated, and clinical details were not presented, nor discussed. Long-lasting effects and other aspects have not been measured in the study. Therefore, I do not think the term “burden” should be specified in the title.

Figure and tables require major revision to my opinion. They should be simplified as a lot of redundant information limits the readers understanding. Table 1 should be checked as some numbers do not add up.

Could the authors explain the use to confidence intervals for their serological data?

I would love to read an improved version of this study with strong serological evidence to prove chikungunya seroprevalence and molecular data to support the coinfection hypothesis.

Author Response

Reviewer 2                                                                                  Comments and Suggestions for Authors

This nigerian-german cross-sectional study was conducted in a large cohort of outpatients from 3 hospitals in distinct regions of Nigeria. After informed consent, patients’ sera were tested on site for malaria antigen detection (detection of ongoing infection) with a rapid diagnostic test. Sera was screened in Germany for chikungunya virus antibodies with a commercial immunoblot kit (detection of past infection). As reported by the authors, results show endemic-like prevalence of chikungunya virus in Nigeria with significant differences in regions and habitation environments. Authors state that the prevalence of malaria infections is lower in contrast to chikungunya infections with significant differences in habitation environments and pregnancy status.

Due to the rarity of arboviral activity in Africa, seroprevalence or molecular data is always welcome for informed readers. This study is particularly unique for its patient cohort sampled in three sites representing a gradient of climates from arid to tropical. This sample collection could deliver valuable data for epidemiologists, clinicians.

The studies’ objective as stated by the authors was to investigate the seroprevalence of chikungunya and determine coinfection with malaria. Overall, in the current state of the manuscript, I do not agree that presented used methods answer the studies objectives and I do not agree with the current conclusions drawn from presented results.  I also feel there is a misuse of certain terms such as “seroprevalence”, “coinfection” and “burden”. Detailed arguments bellow.

To my understanding, a “coinfection” is a simultaneous infection of two pathogens, malaria and chikungunya virus in this instance. However, the methods used detect ongoing infections of malaria and past infections of chikungunya virus. Subsequent results cannot demonstrate simultaneous infections; thus the use of co-infection with the current results is misguiding to my opinion. The authors should consider screening for CHIKV RNA in the sera to prove co-infection.  

ANSWER: We detected  chikungunya and malaria antibodies at the time of sampling, it could be a recent or past infection. In the literature the term “coinfection” is sometimes used when antibodies against 2 different pathogens are present at the same time or when the 2 pathogens are present at the same time. In our study we have an environment where malaria infection happen frequently but where chikungunya infections may reactivate a dormant malaria infection.We cannot say with certainty if the cases are new or old. 

I would also argue that a unique commercial immunoblot result is not sufficient to prove an arboviral infection. In the matter of biological confirmation of arboviral infection, the WHO has established gold standard practices: a positive serological result should be confirmed by a seroneutralisation test (sole confirmatory test available for serological diagnosis) due to cross-reactivity of CHIKV-IgG towards other alphaviruses. Considering those recommendations, the serological evidence present in this study can only indicate a probable CHIKV infection and not a confirmed past CHIKV infection. Limitations concerning cross-reactivity of alphaviruses, prevalence of other alphaviruses in Nigeria, sensitivity and specificity of commercial tests (also, no mention of HRP2-deleted plasmodium in discussion) are all major study limitation which were not discussed. Authors should perform a seroneutralisation assay on a panel of relevant alphaviruses to support their conclusions.

Answer: CHIKV VLP E1&E2 is a highly specific and sensitive(100%) novel molecular biology tool for detecting CHIKV from other alphaviruses (read the references below, sir), as much as PCR; The tool is as good as a neutralizing assay; it was made available to our study by WHO/TDR . We stated have stated in the Limitation section that we did not employ “sero-neutrilization” as a result some of the positives or negatives maybe the opposite. Moreover, sero-neutralization is expensive and laborious especially when working with large samples such as the current sample size employed in this study.

  1. [ref]:https://www.mikrogen.de/english/products/product-overview/weitereinfo/tropical-fever-igg.html]
  2. Pan American Health Organization. Guidelines for Preparedness and Response for Chikungunya Virus Introduction in the Americas; 2011
  3. Pedraza-Escalona M, et al;Isolation and characterization of high affinity and highly stable anti-Chikungunya virus antibodies using ALTHEA Gold Libraries™. BMC Infect Dis. 2021 Oct 30;21(1):1121. doi: 10.1186/s12879-021-06717-0.
  4. Cho B, Jeon BY, Kim J, Noh J, Kim J, Park M, Park S. Expression and evaluation of Chikungunya virus E1 and E2 envelope proteins for serodiagnosis of Chikungunya virus infection. 31;49(5):828-35. doi: 10.3349/ymj.2008.49.5.828.

specificity of commercial tests (also, no mention of HRP2-deleted plasmodium in discussion) are all major study limitation which were not discussed.

Answer: Sorry sir in the present study we were not comparing various RDT commercial diagnostic test kits, In our thinking its not very relevant in this context. Moreso, It is generally recognized that HRP2-based RDTs perform better, especially at low parasite densities, and are more heat-stable than non-HRP2-based RDTs. The current solution to P. falciparum parasite diagnostics is to establish prevalence first, and then determine whether a replacement RDT or microscopy is needed. If HRP2-based P. falciparum-only RDTs are used when a patient is infected solely with parasites lacking HRP2 then a false-negative diagnosis can occur, in this case we made use of  HRP2/pLDH and we were not solely concern or looking out for P.f.alciparum  alone. 

The authors use the term “seroprelavence” for results of chikungunya-IgG detection and malaria antigen detection. However, the term “seroprelavence” is entailed by definition to antibodies detected in a population. Therefore, the direct detection malaria antigen and not the host’s antibodies is “prevalence” rather than “seroprevalence”. The seroprevalence of chikungunya virus and the prevalence of malaria should be treated and compared with more caution. I personally do not believe such data are comparable: on one hand, life long-lasting highly cross-reactive antibodies of an auto-resolutive viral infection are detected and on the other an antigen of chronically infected parasite with complex biological cycle and complex immunological host response.

Answer: Detection of malaria antigen is brought about by an antigen-antibody reaction; seroprevalence in our present context “means detecting antibodies against an antigen in serum or blood samples in the study population”.  Therefore we get exactly what you are saying and the term Seroprevalence or seropositivity can be applied.

When you are referring to prevalence it is the proportion of a particular population found to be affected by a medical condition at a specific time, it could be measured or detected through seroprevalence of seropositivity analysis. The persistence of IgG antibodies lasts for several years according to WHO arboviral tool kit.

In addition, the title of the article mentions a “high burden” of chikungunya virus. However, the burden of a disease is defined as the consequences (health, social aspects and cost to society) of a defined disease or a range of harmful diseases with respect to disabilities in a community. The authors do specify that chikungunya and malaria related symptoms were surveyed. Yet, ongoing chikungunya infection were not evaluated, and clinical details were not presented, nor discussed. Long-lasting effects and other aspects have not been measured in the study. Therefore, I do not think the term “burden” should be specified in the title.

Answer: The term “burden” is used by health economists as you describe it. However, in Public Health it is often used when indicating a high incidence or prevalence of a potentially serious disease such as malaria and chikungunya.

Figure and tables require major revision to my opinion. They should be simplified as a lot of redundant information limits the readers understanding. Table 1 should be checked as some numbers do not add up.

Answer: Table  1 has been reworked, and the others checked for adding up. In the text-only the most important information is given and in the tables all the details are for a more interesting read.

Could the authors explain the use to confidence intervals for their serological data?

Answer: CI because we did a sample survey and the CI95% tells in which interval or range the real population mean lies.

I would love to read an improved version of this study with strong serological evidence to prove chikungunya seroprevalence and molecular data to support the coinfection hypothesis.

Reviewer 3 Report

Mac and colleagues present an elegant and clearly designed study investigating the seroprevalence of chikungunya and malaria in different regions of Nigeria. Overall, I think the paper is well written, the methods are clear and the analysis and conclusions are appropriate for the study population presented. This paper should be well received by the scientific community. 

A few suggestions to improve the paper are as follows

1. pg5, line 160. These abbreviations ABIA, NAS and KAD should be defined here. They are defined later in the paper for Figure 2.

2. It would be useful to clearly define what the authors mean by co-infection. Do you simply mean having serological evidence of chikungunya and malaria infection at the time of sampling? Then there is no confusion as to whether co-infection could represent recent or past infection. 

3. Statistsics that represent significant findings with p values <0.05 should be added to the Figures or in the figure legend and not just in the text. 

4. There should be a definition somewhere in the manuscript as what the authors consider to be "hidden burden". Is this simply morbidity associated with misdiagnosed infections? A social stigma associated with these infections or something completely different? It would also be worth mentioning that hidden burden is difficult to quantify.

5. Discussion - The authors found that chikungunya seroprevalence increased with age. There should be some further discussion here as to whether this was expected relative to what is know about the longevity of chikungunya antibodies following infection. The same could be discussed about malaria seroprevalence. 

6. Limitations - while this section is well written, this section could be expanded to include the significant number of females recruited to the study compared to males. This could lead to bias and also a confounding factor in the analyses performed in the study. Furthermore, there is no discussion during the study about the age distribution of the study population, another possible confounding factor. 

7. There are a few grammatical errors throughout the manuscript. An additional proof-read to correct these would improve clarity and flow. 

Author Response

Reviewer 3

Comments and Suggestions for Authors

Mac and colleagues present an elegant and clearly designed study investigating the seroprevalence of chikungunya and malaria in different regions of Nigeria. Overall, I think the paper is well written, the methods are clear and the analysis and conclusions are appropriate for the study population presented. This paper should be well received by the scientific community. 

A few suggestions to improve the paper are as follows

  1. pg5, line 160. These abbreviations ABIA, NAS and KAD should be defined here. They are defined later in the paper for Figure 2.

Answer: Corrected sir, thanks for your observation.

  1. It would be useful to clearly define what the authors mean by co-infection. Do you simply mean having serological evidence of chikungunya and malaria infection at the time of sampling? Then there is no confusion as to whether co-infection could represent recent or past infection.

Answer:  we simply meant having serological evidence of chikungunya and malaria infection at the time of sampling. In the literature the term “coinfection” is sometimes used when antibodies against 2 different pathogens are present at the same time or when the 2 pathogens are present at the same time. In our study we have an environment where malaria infection happens frequently but where chikungunya infections may reactivate a dormant malaria infection. Many thanks sir.

Statistsics that represent significant findings with p values <0.05 should be added to the Figures or in the figure legend and not just in the text. 

Answer: Corrected, many thanks for your observation and suggestion(s).

  1. There should be a definition somewhere in the manuscript as what the authors consider to be "hidden burden". Is this simply morbidity associated with misdiagnosed infections? A social stigma associated with these infections or something completely different? It would also be worth mentioning that hidden burden is difficult to quantify.

Answer: Burden in Public Health is often synonymous with high incidence or high prevalence of a potentially serious disease (Health economists have a more complex definition). “Hidden” means undetected or unrecognized by the health services or the public.

  1. Discussion - The authors found that chikungunya seroprevalence increased with age. There should be some further discussion here as to whether this was expected relative to what is know about the longevity of chikungunya antibodies following infection. The same could be discussed about malaria seroprevalence. 

Answer: It is an established fact that CHIKV increases with age, and the antibodies may persist for many years (sometimes 10 years and longer [WHO arboviral tool Kit].

  1. Limitations - while this section is well written, this section could be expanded to include the significant number of females recruited to the study compared to males. This could lead to bias and also a confounding factor in the analyses performed in the study. Furthermore, there is no discussion during the study about the age distribution of the study population, another possible confounding factor. 

Answer: Thank you for your observation and suggestion, it has been included as suggested.

  1. There are a few grammatical errors throughout the manuscript. An additional proof-read to correct these would rove clarity and flow.

Answer: Rectified.

Many thanks for your valuable input, suggestion and time sir.

Round 2

Reviewer 1 Report

Dear authors

Although I still think there is unnecessary data, as you have pointed out, this does not affect the main results.

I consider it to be a publishable article.

Author Response

Reviewer 1

Comments and Suggestions for Authors

TITLE

Avoid that the title is gimmicky. I suggest removing the words "high" and "endemicity"

Answer: Sir it what we saw that made us to come up with the tittle. According to your suggestion we shall take out the “High”.

New tittle: The seroprevalence and hidden burden of chikungunya endemicity and malaria mono and coinfection in Nigeria”

ABSTRACT

I suggest rewriting the methods section so that it is better understood. It should be indicated if the sample was collected from healthy volunteers or if some type of sampling was done.

Answer: The abstract method section was rectified regarding  sampling method and from whom the samples were  collected.  

BACKGROUND

The sentences between lines 43-47 do not add context to the study. I suggest increasing the context regarding the seroprevalence of both diseases in the continent or the country. I suggest considering this reference: Igbasi U, Oyibo W. Seroprevalence of immunoglobulin G and E among out-patients with malaria in Ikorodu Lga, Lagos, Nigeria. Microbes Infect Chemother. 2022; 2: e1376.

Answer: Rectified with [35]

METHODS

Support the reason for including such young children in the study (0 months). How frequent is the prevalence in this age? How they have evaluated the bias of decreasing prevalence by having children in the study.

Answer: The  burden of CHIKV infection is particularly in the very young age groups [arboviral took KIT, 2018]. Most  frequently the infection is diagnosed as malaria or other febrile illness.  [WHO arboviral tool KIT, 2018].  

I strongly suggest removing children under 5 years of age from the analysis, due to the high bias they cause, not only in seroprevalence but also in the symptoms and signs evaluated.

Answer: As mentioned above under-fives are particularly affected by arbovirus diseases and malaria. [WHO, Dengue tool, KIT]. This is why we included them in the current study. And we think we had important findings Therefore we think it is important to include them showing that many infections occur at such a low age.  

Because they have chosen blood bank samples. Prior to the taking, an interrogation is made to define if the donation proceeds.

Answer: Blood donors are nor checked for Antibodies in our target diseases. This was a small additional study which did not change our overall results.

This affects serology results for both diseases and even more so for malaria. How have clinical questions been evaluated in blood bank samples with clinical histories?

Answer: All the blood samples from the blood bank had all the medical data such as date, blood group, genotype, matrix tracking code, donor ID number or product code (Which contain the age and sex, entered in hospital records) expiration information. We had all the information. The samples were selected randomly by the hospital laboratory technician of participating hospitals. 

How much missing data did you have in those 110 samples? How did you choose which samples to obtain from the blood bank? Was it also by simple random sampling?

Answer: All the blood samples from the blood bank had all the medical data such as date, blood group, genotype, matrix tracking code, donor ID number, product code (Which contain the age and sex, entered in hospital records, very confidential) expiration information. We had all the information. It was selected randomly by the hospital laboratory technician located in the hospitals.

RESULTS

Avoid repeating the data shown in the tables in the text.

Answer: In the text we put only the important data without quoting the 95%Cis and p-values. These the reader can find in the text.

Increase the size of figure 2.

Answer: The size has been increased.

Figure 3 can be described in the text, it is unnecessary.

Answer: Taken out

Figures 4 and 5 can be shown in a table.

Answer: Taken out as suggested.

DISCUSSION

Consider the discussion in other contexts of the disease. I consider this research relevant in terms of socioeconomic aspects: Olivera MJ, Peña C, Yasnot MF, Padilla J. Socioeconomic determinants for malaria transmission risk in Colombia: An ecological study. Microbes Infect Chemother. 2022; 2: e1339

Answer: Noted sir and taken into account (see text)

It is an interesting study, however, the wide age range and the use of blood bank samples lead to serious deficiencies. I suggest removing it from the analysis.

Answer: Answer: Various arboviral infections, such as chikungunya, have been reported to be transmitted through blood donation. Blood bank samples are crucial for studies like this. In most blood banks and clinics, arboviruses are rarely screened [Haliya et al [2021]: Seroprevalence of Dengue and Chikungunya antibodies among blood donors in Dar es Salaam and Zanzibar]. Due to inadequate molecular diagnostic tools and maternal immunity, juveniles and neonates bear the hidden burden of misdiagnosis and associated morbidities of arboviruses[WHO, dengue took KIT].  We do not know why it should be ruled out or removed in the current study, sir. For us it’s the very core part of the substance and content of the study.

Reviewer 2 Report

The authors do not answer any of my concerns and confirm confusion with basic notions of used tests.  

Author Response

Answers and rebuttals .

This nigerian-german cross-sectional study was conducted in a large cohort of outpatients from 3 hospitals in distinct regions of Nigeria. After informed consent, patients’ sera were tested on site for malaria antigen detection (detection of ongoing infection) with a rapid diagnostic test. Sera was screened in Germany for chikungunya virus antibodies with a commercial immunoblot kit (detection of past infection). As reported by the authors, results show endemic-like prevalence of chikungunya virus in Nigeria with significant differences in regions and habitation environments. Authors state that the prevalence of malaria infections is lower in contrast to chikungunya infections with significant differences in habitation environments and pregnancy status.

Due to the rarity of arboviral activity in Africa, seroprevalence or molecular data is always welcome for informed readers. This study is particularly unique for its patient cohort sampled in three sites representing a gradient of climates from arid to tropical. This sample collection could deliver valuable data for epidemiologists, clinicians.

The studies’ objective as stated by the authors was to investigate the seroprevalence of chikungunya and determine coinfection with malaria. Overall, in the current state of the manuscript, I do not agree that presented used methods answer the studies objectives and I do not agree with the current conclusions drawn from presented results.  I also feel there is a misuse of certain terms such as “seroprevalence”, “coinfection” and “burden”. Detailed arguments bellow.

To my understanding, a “coinfection” is a simultaneous infection of two pathogens, malaria and chikungunya virus in this instance. However, the methods used detect ongoing infections of malaria and past infections of chikungunya virus. Subsequent results cannot demonstrate simultaneous infections; thus the use of co-infection with the current results is misguiding to my opinion. The authors should consider screening for CHIKV RNA in the sera to prove co-infection.  

ANSWER: In the literature the term “coinfection” is sometimes used when antibodies against 2 different pathogens are present at the same time or when the 2 pathogens are present at the same time. In the present study “we simply meant having serological evidence of chikungunya and malaria infection at the time of sampling”. It could be a recent or past infection. In our study we have an environment where malaria infection happens frequently but where chikungunya infections may reactivate a dormant malaria infection. We cannot say with certainty if the cases are new or old. 

I would also argue that a unique commercial immunoblot result is not sufficient to prove an arboviral infection. In the matter of biological confirmation of arboviral infection, the WHO has established gold standard practices: a positive serological result should be confirmed by a seroneutralisation test (sole confirmatory test available for serological diagnosis) due to cross-reactivity of CHIKV-IgG towards other alphaviruses. Considering those recommendations, the serological evidence present in this study can only indicate a probable CHIKV infection and not a confirmed past CHIKV infection. Limitations concerning cross-reactivity of alphaviruses, prevalence of other alphaviruses in Nigeria, sensitivity and specificity of commercial tests (also, no mention of HRP2-deleted plasmodium in discussion) are all major study limitation which were not discussed. Authors should perform a seroneutralisation assay on a panel of relevant alphaviruses to support their conclusions.

Answer: CHIKV VLP E1&E2 is a highly specific and sensitive(100%) novel molecular biology tool that detect CHIKV from other alphaviruses (read the references below, sir), as much as PCR; The tool is as good as a neutralizing assay; it was made available to our study by WHO/TDR . We have stated in the limitation section of the manuscript that not employ “sero-neutrilization was a limitation. As a result some of the positives or negatives maybe the “false”.

specificity of commercial tests (also, no mention of HRP2-deleted plasmodium in discussion) are all major study limitation which were not discussed.

Answer:  We were not comparing various RDT commercial diagnostic test kits, as a result discussing RDT in the discussion as a limitation was not very relevant in this context. The study was mainly concern with serology of the two vectors. In our thinking its not. Moreso, It is generally recognized that HRP2-based RDTs perform better, especially at low parasite densities, and are more heat-stable than non-HRP2-based RDTs. The current solution to P. falciparum parasite diagnostics is to establish prevalence first, and then determine whether a replacement RDT or microscopy is needed. If HRP2-based P. falciparum-only RDTs are used when a patient is infected solely with parasites lacking HRP2 then a false-negative diagnosis can occur, in this case we made use of  HRP2/pLDH and we were not solely concern or looking out for P.f.alciparum  alone. 

The authors use the term “seroprelavence” for results of chikungunya-IgG detection and malaria antigen detection. However, the term “seroprelavence” is entailed by definition to antibodies detected in a population. Therefore, the direct detection malaria antigen and not the host’s antibodies is “prevalence” rather than “seroprevalence”. The seroprevalence of chikungunya virus and the prevalence of malaria should be treated and compared with more caution. I personally do not believe such data are comparable: on one hand, life long-lasting highly cross-reactive antibodies of an auto-resolutive viral infection are detected and on the other an antigen of chronically infection parasite with complex biological cycle and complex immunological host response.

Answer: Detection of malaria antigen is brought about by an antigen-antibody reaction; seroprevalence in our present context “means detecting antibodies against antigen in serum or blood samples in the study population”. Therefore, we get exactly what you are saying and the term Seroprevalence or seropositivity can be applied.

When you are referring to prevalence it is the proportion of a particular population found to be affected by a medical condition at a specific time, it could be measured or detected through seroprevalence of seropositivity analysis. The persistence of IgG antibodies lasts for several years according to WHO arboviral tool kit.

In addition, the title of the article mentions a “high burden” of chikungunya virus. However, the burden of a disease is defined as the consequences (health, social aspects and cost to society) of a defined disease or a range of harmful diseases with respect to disabilities in a community. The authors do specify that chikungunya and malaria related symptoms were surveyed. Yet, ongoing chikungunya infection were not evaluated, and clinical details were not presented, nor discussed. Long-lasting effects and other aspects have not been measured in the study. Therefore, I do not think the term “burden” should be specified in the title.

Answer: The term “burden” is used by health economists as you are describe it. However, in Public Health it is often used when indicating a high incidence or prevalence of a potentially serious disease as malaria and chikungunya.

Figure and tables require major revision to my opinion. They should be simplified as a lot of redundant information limits the readers understanding. Table 1 should be checked as some numbers do not add up.

Answer: Table  1 has been reworked, and the others checked for adding up. In the text only the most important information is given and in the tables all the details for more interested read.

Could the authors explain the use to confidence intervals for their serological data?

Answer: CI because we did a sample survey and the CI 95% tells in which interval or range the real population mean lies.

I would love to read an improved version of this study with strong serological evidence to prove chikungunya seroprevalence and molecular data to support the coinfection hypothesis.

Answer: We believe we have answered all of your suggestions and comments in a candid and fair manner. Where possible, we have also made the necessary changes. We greatly appreciate your input, suggestions, and time taken to review our manuscript.
